Accepted at the ICLR 2024 Workshop on AI4Differential Equations In Science

# MULTI-LATTICE SAMPLING OF QUANTUM FIELD THEORIES VIA NEURAL OPERATOR-BASED FLOWS

**Bálint Máté & François Fleuret**
University of Geneva
{balint.mate,francois.fleuret}@unige.ch

## ABSTRACT

We consider the problem of sampling discrete field configurations $\phi$ from the Boltzmann distribution $[d\phi]Z_1^{-1}e^{-S_1[\phi]}$, where $S_1$ is the lattice-discretization of the continuous Euclidean action $\mathcal{S}_1$ of some quantum field theory. Since such densities arise as the approximation of the underlying functional density $[\mathcal{D}\phi(x)]\mathcal{Z}_1^{-1}e^{-\mathcal{S}_1[\phi(x)]}$, we frame the task as an instance of operator learning. In particular, we propose to approximate a time-dependent operator $\mathcal{V}_t$ whose time integral provides a mapping between the functional distributions of the free theory $[\mathcal{D}\phi(x)]\mathcal{Z}_0^{-1}e^{-\mathcal{S}_0[\phi(x)]}$ and of the target theory $[\mathcal{D}\phi(x)]\mathcal{Z}_1^{-1}e^{-\mathcal{S}_1[\phi(x)]}$. Once a particular lattice is chosen, the operator $\mathcal{V}_t$ can be discretized to a finite dimensional, time-dependent vector field $V_t$ which in turn induces a continuous normalizing flow between finite dimensional distributions over the chosen lattice. This flow can then be trained to be a diffeormorphism between the discretized free and target theories $[d\phi]Z_0^{-1}e^{-S_0[\phi]}$, $[d\phi]Z_1^{-1}e^{-S_1[\phi]}$. We run experiments on the 2-dimensional $\phi^4$-theory to explore to what extent such operator-based flow architectures generalize to lattice sizes they were not trained on and show that pre-training on smaller lattices can lead to speedup over training directly on the target lattice size.

## 1 INTRODUCTION

Consider $\mathcal{S}$ as an action characterizing a quantum field theory, with $S$ representing its discretization to a lattice. Albergo et al. (2019) suggest a method for sampling from the lattice quantum field theory described by $S$. This involves a normalizing flow parameterizing a density function $q_\theta$ of discrete fields over the lattice, and optimizing the parameters $\theta$ until $q_\theta$ closely approximates the probability density $\frac{e^{-S}}{Z}$, where $Z$ is the normalizing constant of $e^{-S}$.

Operator learning promotes the viewpoint that the lattice/mesh is merely a computational tool, and model should capture the underlying continuous physics. Kovachki et al. (2021) term this property of models discretization invariance. [1] In this work, we apply the same idea to the task of sampling from lattice quantum field theories, motivated by the fact that lattice field theories also emerge as the discretization of continuous field theories.

Suppose now that the field theory is defined on some domain $D$. Once a lattice, as a discretization of $D$, is chosen, one can construct a continuous normalizing flow Chen et al. (2018) as a time-dependent vector field $V_t$ that parametrizes the direction along which probability mass moves. Generalizing this idea, we propose to parametrize a time-dependent operator $\mathcal{V}_t$ from the space of functions on $D$ to itself that defines the direction in which functional probability mass moves. Such an operator can then be used to map the functional distributions $[\mathcal{D}\phi(x)]\mathcal{Z}_0^{-1}e^{-\mathcal{S}_0[\phi(x)]}$, $[\mathcal{D}\phi(x)]\mathcal{Z}_1^{-1}e^{-\mathcal{S}_1[\phi(x)]}$ to one another. Computationally the operator $\mathcal{V}_t$ can only be accessed by a choice of a lattice which induces a vector field $V_t$ as the discretization of $\mathcal{V}_t$.

---

[1]Discretization invariance means that the neural operator evaluated on finer and finer discretizations approximates the continuous operator. Thus, strictly speaking, it is not a requirement of invariance rather that of convergence.

We then train this vector field to be a diffeomorphism between the discretized free and target theories, $[d\phi]Z_0^{-1}e^{-S_0[\phi]}$ and $[d\phi]Z_1^{-1}e^{-S_1[\phi]}$. The upside of using a operator-based flow will be that a single model can be used to operate on multiple discretizations of the same underlying continuous system. Figure 1 provides a schematic overview of the objects and their relation in this paragraph.

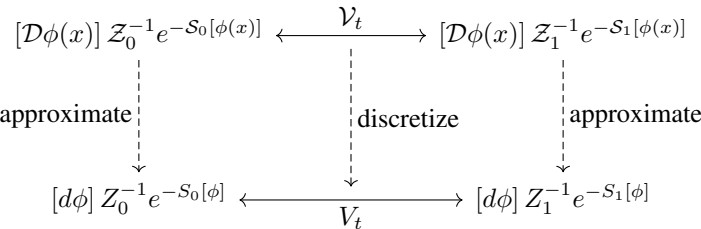

Figure 1: Schematic overview of the probability distributions of interest. The top row shows the functional distributions of the free theory and the target theory connected by the time dependent operator $\mathcal{V}_t$. Descending to the bottom row means approximating all the objects of the top row on a lattice. In particular, in the bottom row all objects are finite dimensional, well-defined and can be numerically worked with.

The structure of the paper is as follows:

- Section §2 introduces the relevant background on the $\phi^4$ quantum field theory.

- In Section §3 we describe an operator-based normalizing flow architecture that we used in our experiments.

- Section §4 documents our experiment on the $\phi^4$ theory.

- Section §A contains the relevant background on continuous normalizing flows, Boltzmann densities and neural operators.

## 2 The $\phi^4$ (Lattice) Quantum Field Theory[2]

Let us now consider the Euclidean action on real valued scalar fields $\phi(x)$ with periodic boundary conditions on the $D$-dimensional hypercube of edge length $L$, $\phi : (\mathbb{R}/L\mathbb{Z})^D \to \mathbb{R}$, for some constants $m^2$ and $g$

$$\mathcal{S}[\phi] = \int_{(\mathbb{R}/L\mathbb{Z})^D} d^D x \left[ (\nabla \phi)^2 + m^2 \phi^2 + g\phi^4 \right] \tag{1}$$

where we dropped the argument $x$ of the field $\phi(x)$ for notational convenience. To estimate the expectation value of an observable $\mathcal{O}$, we need to average over all field configurations that satisfy the boundary conditions, with each configuration weighted by the exponential of the negative action

$$\langle \mathcal{O} \rangle = \frac{\int \mathcal{D}\phi \, \mathcal{O}[\phi] e^{-\mathcal{S}[\phi]}}{\int \mathcal{D}\phi \, e^{-\mathcal{S}[\phi]}} \tag{2}$$

The action $\mathcal{S}$ corresponds to the energy function $f$ of a Boltzmann density and the denominator $\mathcal{Z} = \int \mathcal{D}\phi \, e^{-\mathcal{S}[\phi]}$ to the normalizing constant as introduced in Appendix §A.1.

Equations (13) and (14) describe an infinite dimensional system. To tackle it numerically, one first needs to discretize it to a lattice. This comes at the cost of losing the information contained in the high-frequency components as the highest possible frequency of a periodic function on a lattice with edge length $L$ with $N$ nodes is $\frac{2\pi N}{L}$. The hope is that one can do the same on larger and larger lattices, and as the lattice approaches the continuum limit, the error due to discretization converges to zero.

---

[2]We recommend the book (Thijssen, 2007, Chapter 15) for further details on lattice field theories.

DISCRETE REPRESENTATIONS ON LATTICES

To discretise the action, we consider fields living on the points located at $\left\{ \frac{0}{N}, \frac{L}{N}, ..., \frac{(N-1)L}{N} \right\}^d$ forming a periodic lattice with cardinality $N^D$ and lattice spacing $a = L/N$. We then turn integrals into sums $\int_{(\mathbb{R}/L\mathbb{Z})^D} d^D x \to a^D \sum_x$ and differentials into differences between nearest neighbors $\partial_i \phi \to \frac{1}{a} \phi(x + \mu_i) - \phi(x)$, where $\mu_i$ is the generator of lattice along the $i-$th coordinate axis. After these substitutions we end up with the following discretised action on the lattice,

$$ S[\phi] = a^D \left\{ \frac{1}{a^2} \sum_{x,\mu} (\phi_{x+\mu} - \phi_x)^2 + \sum_x m^2 \phi_x^2 + g\phi_x^4 \right\} \tag{3} $$

where $x$ runs over the lattice sites and $\mu$ over the generators of the lattice.

## 3 FLOWS PARAMETRISED BY NEURAL OPERATORS

Let now $\phi \in \mathbb{R}^{N \times \cdots \times N}$ be a discretized scalar field on a lattice. The architecture then consists of the following sequence of steps, where the subscript $\theta$ denotes trainable parameters,

1. Use a per-node neural network $f_\theta$ to embed the field values, $\phi_{emb} = f_\theta(\phi) \in \mathbb{R}^{c \times N \times \cdots \times N}$.
2. Use a neural network to parametrize $c$-many continous spherically symmetric kernels $K_\theta(r)$. Let then $\tilde{K}_\theta$ be the evaluation of the continous kernels on the lattice.
3. Mask out the origin of the discrete kernel, i.e. set $\tilde{K}_\theta[:, \mathbf{0}] = 0$.
4. Perform a the channel-wise convolution $\phi_{emb} \star \tilde{K}_\theta$ and denote the result by $C \in \mathbb{R}^{c \times N \times \cdots \times N}$. Because of the previous step, $C_i$ is independent of $\phi_i$, and we will call it the conditioner (Chen & Duvenaud (2019)).
5. Apply a per-node neural network $\tau_\theta$ to the concatenation $(C, \phi_{emb})$ with output $Y = \tau_\theta(C, \phi_{emb}) \in \mathbb{R}^{T \times N \times \cdots \times N}$.
6. Contract the first dimension of $Y$ with a vector of length $T$ that only depends on time.
7. Finally, denoting all the above steps as $i$, we set the output of the model to be $V(\phi, t) = \frac{1}{2} * (i(\phi, t) - i(-\phi, t))$. This enforces the $\mathbb{Z}_2$ symmetry of the system.

To compute the divergence of the architecture one needs the Jacobians of the per-point operations $f_\theta$ and $\tau_\theta$, $K_\theta$ does not have to be differentiated through.

THE FREE THEORY AS AN INITIAL DENSITY

The normalizing flow architecture described in Section §A.1 requires an initial density from which samples can easily be drawn. Instead of sampling from a standard gaussian at every node, we choose a more physical initial density by setting $g = 0$ in the action (17). This results in the free theory with a gaussian Boltzmann density that becomes diagonal in momentum space. Position and momentum space are related by a discrete Fourier transform

$$ \phi_x = \frac{1}{\sqrt{N^D}} \sum_p \tilde{\phi}_p e^{i 2\pi \langle p, x \rangle} \qquad\qquad \tilde{\phi}_p = \frac{1}{\sqrt{N^D}} \sum_x \phi_x e^{-i 2\pi \langle p, x \rangle} \tag{4} $$

where $p$ runs over $\left\{ -\frac{\lfloor N/2 \rfloor}{L}, ...., \frac{0}{L}, ..., \frac{\lceil (N-1)/2 \rceil}{L} \right\}^D$ and the prefactor $\frac{1}{\sqrt{N^D}}$ makes the map $\{\phi_x\} \leftrightarrow \{\tilde{\phi}_p\}$ unitary. The covariance matrix of the free theory is diagonalized in the momentum basis $\frac{1}{\sqrt{N^D}} e^{2\pi i \langle x, p \rangle}$ with eigenvalues

$$ S\left[ \frac{1}{\sqrt{N^D}} e^{2\pi i \langle x, p \rangle} \right] = a^D \left( m^2 + \frac{1}{a^2} \sum_\mu 2 - 2\cos(2\pi p_\mu a) \right) \tag{5} $$

To constrain the sampling to real valued fields, we sample $p$ from the hermitian symmetric subspace of *real dimension $N^D$* of the Fourier-space of *complex dimension $N^D$*.

## 4 EXPERIMENTS

One of the core dilemmas when choosing the lattice size lies in balancing the cost efficiency of a smaller lattice against the better approximation of the underlying continuous system of a larger one. Ideally, one would train on a small lattice and evaluate on a large one, but it is unreasonable to expect this approach to work well as the smaller lattices cannot capture the higher frequency components of the system. We thus explore the next best strategy: extensive pretraining on a small lattice followed by transferring the model to a larger lattice size and fine-tuning it with a reduced number of steps.

### 4.1 MULTI-LATTICE SAMPLING IN $D = 1$ DIMENSION

We now work in $D = 1$ dimensions. Strictly speaking, a one dimensional lattice does not correspond to a quantum field theory, rather it describes the trajectory of a quantum mechanical particle in a potential. Nonetheless, it's the simplest setup in which we can experiment and serves as a good starting point. We also fix $L = 4, m^2 = -4, g = 1$ and train a single model for $5000$ steps with mesh size uniformly sampled at each training step from $N = L/a \in \{4, 8, 16, ..., 128\}$. We then evaluate performance on lattices of size $N = L/a$ up to $512$ by sampling from the trained model to calculate the effective sample size

$$ESS = \left( \frac{1}{N} \sum_i w_i \right)^2 \bigg/ \left( \frac{1}{N} \sum_i w_i^2 \right) \tag{6}$$

where $w_i$ is the importance weight $p(\phi_i)/q_\theta(\phi_i)$. We also estimate the two-point correlation function

$$G(x, y)[\phi] := \phi(x)\phi(y). \tag{7}$$

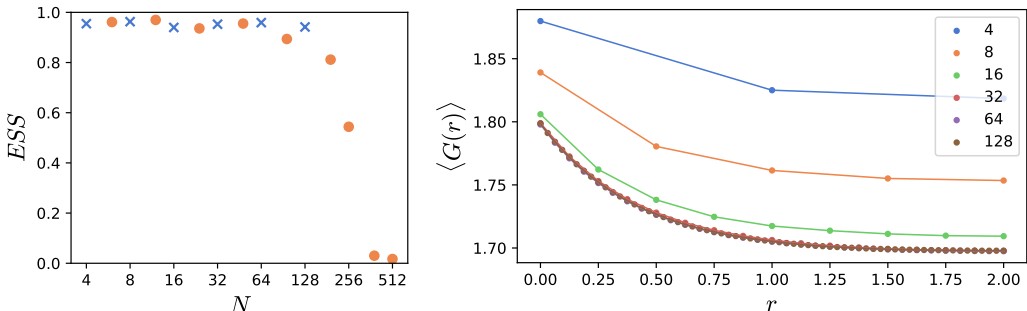

Figure 2: Experiment 4.1. (**Left**) $ESS$ computed from $16384$ samples at different lattice sizes. The blue crosses correspond to lattice sizes that the model was trained on, while orange dots denote lattice sizes unseen by the network during training. (**Right**) The two-point correlation function $G(x, y)$ computed from $16384$ samples on different lattice sizes the model was trained on. Because of the symmetries of the task the correlation function only depends on the distance $r = |x - y|$, thus the function $G(r)$ is plotted.

### 4.2 PRETRAINING ON SMALLER LATTICES

In this experiment we consider the target $D = 2, L = 12, m^2 = -4, g = 5.276, N = 64$. Instead of training directly on the $N = 64$ lattice, we pretrain on a sequence of smaller lattices as they are significantly cheaper to work on. We start training on a $12 \times 12$ lattice for $2000$ steps, after which we train on lattices of size $16^2, 20^2, 24^2, 28^2, 32^2, 36^2, 40^2, 44^2, 48^2, 52^2, 56^2, 60^2$ for $250$ training steps each. Finally, we train on the target size $64 \times 64$ for $1000$ steps. As a baseline, we also train the same architecture only on the target size for the same total number of steps ($6000$).

While the performance, as measured by the effective sample size on target lattice, is comparable after training (Table 1), the training procedure that "trained through" the smaller lattices was $\sim 2.4$-times quicker to train(Figure 3). Figure 4 shows the estimated correlation function at different lattice

sizes computed from model checkpoints saved right after taking the last training step on the given lattice size.

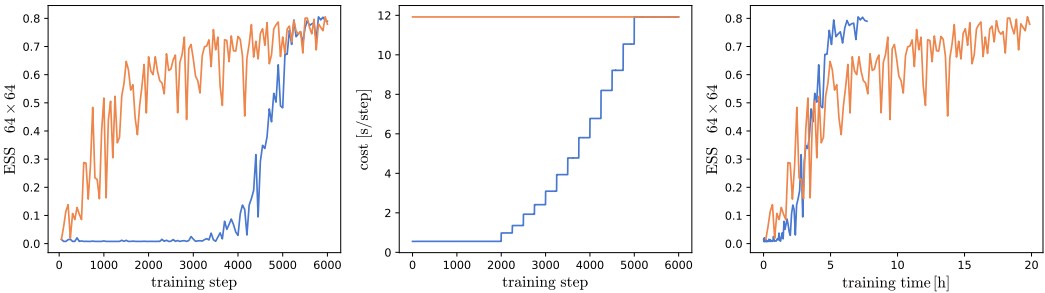

Figure 3: Experiment 4.2. $ESS$ estimated during training on $128$ samples plotted against the number of training steps (left) and training time (right). Time required to take a single step (center). All plots contain two curves, one for the model that is trained on the sequence of increasing lattice sizes (blue) and one that is only trained on the $64 \times 64$ lattice (orange).

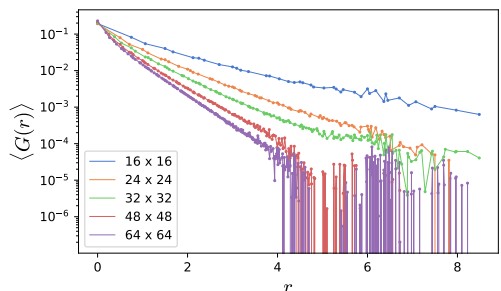

Figure 4: Experiment 4.2. The two-point correlation function $G(x, y)$ computed from $16384$ samples with (right) and without (left) log-scaled $y$-axis. These curves are computed from model checkpoints saved right after taking the last training step on the given lattice size. Because of the symmetries of the task the correlation function only depends on the distance $r = |x - y|$, thus the function $G(r)$ is plotted.

Table 1: Experiment 4.2. ESS values on $16384$ samples from the trained model. Since training on larger lattices degrades performance on smaller ones (Figure 6), the model is evaluated directly after the last training step has been performed on a given lattice size. The final column marked with $\flat$ denotes the baseline model.

| $N \times N$ | $16 \times 16$ | $24 \times 24$ | $32 \times 32$ | $48 \times 48$ | $64 \times 64$ | $64 \times 64^{\flat}$ |
|---|---|---|---|---|---|---|
| ESS | 0.8937 | 0.8628 | 0.8771 | 0.7736 | 0.7824 | 0.7722 |

## 5 CONCLUSION

In this work we explored the idea of using operator-based normalizing flows for sampling from the $\phi^4$ quantum field theory. Experiment 4.1 showed that models trained on a collection of lattices do not generalize zero-shot to lattice sizes much larger than those of the training set. They do generalize with a reasonable performance to lattice sizes slightly larger than the ones it has been trained on. Making use of this observation, in experiment 4.2 we show that training a model on a sequence of meshes of increasing size leads to faster training compared to training directly on the target lattice size.

## 6 ACKNOWLEDGEMENT

The authors were supported by the Swiss National Science Foundation under grant number CR-SII5_193716 - "Robust Deep Density Models for High-Energy Particle Physics and Solar Flare Analysis (RODEM)". We thank Samuel Klein for discussions.

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

## A  BACKGROUND

### A.1  CONTINUOUS NORMALIZING FLOWS

A continuous normalizing flow Chen et al. (2018) is a density estimator that operates by pushing forward a simple, usually Gaussian, initial density $q_0$ along a parametric, time-dependent vector field $V_\theta : [0,1] \times \mathbb{R}^n \to \mathbb{R}^n$. Explicitly, the pushforward density $q_\theta$ is given by

$$\log q_\theta(x_1) = \log q_0(x_0) + \int_1^0 dt\, \nabla \cdot V_\theta(t, x_t) \tag{8}$$

where $\nabla$ is the divergence operator in the spatial coordinates and $x_t$ is the integral curve of $V_\theta$ that passes through $x_1$ at $t = 1$. In this work all normalizing flows will be continuous normalizing flows, and we will refer to them as normalizing flows or even just flows for brevity.

**Boltzmann distributions**  The Boltzmann distribution of an energy function[3] $f : \mathbb{R}^n \to \mathbb{R}$ is a probability distribution with density function

$$p(x) = \frac{1}{Z} e^{-f(x)} \tag{9}$$

where $Z = \int dx\, e^{-f(x)}$ is the normalizing constant ensuring that the density function integrates to 1. Boltzmann distributions appear in the context of the canonical ensemble, a statistical ensemble that describes a system in thermal equilibrium with an external heat reservoir. Such Boltzmann distributions describe the molecular systems in thermal equilibrium as well as Wick-rotated quantum field theories. Learning to sample from Boltzmann distributions using only the energy function (i.e. without true samples) can be done by training a normalizing flowAlbergo et al. (2019; 2021a; 2022); Abbott et al. (2022); Albergo et al. (2021b); Boyda et al. (2021); Noé et al. (2018); Köhler et al. (2020); Nicoli et al. (2020; 2021; 2023); de Haan et al. (2021); Gerdes et al. (2022); Máté & Fleuret (2023), usually, to minimize the reverse KL divergence

$$KL[q_\theta, p] = \mathbb{E}_{x \sim q_\theta} \left[ \log q_\theta(x) - \log p(x) \right] \tag{10}$$
$$= \mathbb{E}_{x \sim q_\theta} \left[ \log q_\theta(x) + f(x) \right] + Z \tag{11}$$

where $q_\theta$ is the density realized by the normalizing flow (8). Once a density $q_\theta$, approximating $p = Z^{-1}e^{-f}$, is learnt, one can use importance sampling to correct for small inaccuracies of $q_\theta$ when estimating the expected value of an observable $\mathcal{O}$

$$\langle \mathcal{O} \rangle := \mathbb{E}_{x \sim p} \left[ \mathcal{O}(x) \right] = \mathbb{E}_{x \sim q_\theta} \left[ \mathcal{O}(x) \frac{p(x)}{q_\theta(x)} \right] \tag{12}$$

---

[3]Assuming that $\exp(-f)$ is integrable.

## A.2  The $\phi^4$ (Lattice) Quantum Field Theory[4]

Let us now consider the Euclidean action on real valued scalar fields $\phi(x)$ with periodic boundary conditions on the $D$-dimensional hypercube of edge length $L$, $\phi : (\mathbb{R}/L\mathbb{Z})^D \to \mathbb{R}$, for some constants $m^2$ and $g$

$$\mathcal{S}[\phi] = \int_{(\mathbb{R}/L\mathbb{Z})^D} d^D x \left[ (\nabla \phi)^2 + m^2 \phi^2 + g\phi^4 \right] \tag{13}$$

where we dropped the argument $x$ of the field $\phi(x)$ for notational convenience. To estimate the expectation value of an observable $\mathcal{O}$, we need to average over all field configurations that satisfy the boundary conditions, with each configuration weighted by the exponential of the negative action

$$\langle \mathcal{O} \rangle = \frac{\int \mathcal{D}\phi \, \mathcal{O}[\phi] e^{-\mathcal{S}[\phi]}}{\int \mathcal{D}\phi \, e^{-\mathcal{S}[\phi]}} \tag{14}$$

The action $\mathcal{S}$ corresponds to the energy function $f$ of a Boltzmann density and the denominator $\mathcal{Z} = \int \mathcal{D}\phi \, e^{-\mathcal{S}[\phi]}$ to the normalizing constant as introduced in Section §A.1.

Equations (13) and (14) describe an infinite dimensional system. To tackle it numerically, one first needs to discretize it to a lattice. This comes at the cost of losing the information contained in the high-frequency components as the highest possible frequency of a periodic function on a lattice with edge length $L$ with $N$ nodes is $\frac{2\pi N}{L}$. The hope is that one can do the same on larger and larger lattices, and as the lattice approaches the continuum limit, the error due to discretization converges to zero.

### Discrete representations on lattices

To discretise the action, we consider fields living on the points located at $\left\{ \frac{0}{N}, \frac{L}{N}, ..., \frac{(N-1)L}{N} \right\}^d$ forming a periodic lattice with cardinality $N^D$ and lattice spacing $a = L/N$. We then turn integrals into sums and differentials into differences between nearest neighbors

$$\partial_i \phi \to \frac{1}{a} \phi(x + \mu_i) - \phi(x) \tag{15}$$

$$\int_{(\mathbb{R}/L\mathbb{Z})^D} d^D x \to a^D \sum_x \tag{16}$$

where $\mu_i$ denotes the generator of the lattice along the $i-$th coordinate axis. After these substitutions we end up with the following discretised action on the lattice,

$$S[\phi] = a^D \left\{ \frac{1}{a^2} \sum_{x,\mu} (\phi_{x+\mu} - \phi_x)^2 + \sum_x m^2 \phi_x^2 + g\phi_x^4 \right\} \tag{17}$$

where $x$ runs over the lattice sites and $\mu$ over the generators of the lattice. It is customary to absorb all the occurrences of $a$ in the above formula by rescaling $\phi$

$$\phi \to a^{D/2-1}\phi, \qquad m \to am, \qquad g \to a^{4-D}g \tag{18}$$

This results in an alternative form of the action

$$S[\phi] = \sum_{x,\mu} (\phi_{x+\mu} - \phi_x)^2 + \sum_x m^2 \phi_x^2 + g\phi_x^4 \tag{19}$$

While the action (19) has the advantage of not being dependent on the lattice spacing $a$, we will continue working with (17) keeping the relation between different lattice sizes and to the underlying continuous setting explicit.

---

[4]We recommend the book (Thijssen, 2007, Chapter 15) for further details on lattice field theories.

## A.3 NEURAL OPERATORS

Neural Operators Kovachki et al. (2021) are trainable function-to-function mappings. In particular, both their domains and codomains are infinite dimensional function spaces. In practice, one works with neural operators by choosing a mesh/lattice $X \subset \mathbb{R}^n$, representing functions by their evaluations on $X$ and let the neural operator operate on this collection of evaluations. By design, neural operators can be evaluated on lattices of different size. Importantly, if a neural operator is applied to a sequence of meshes $X_i$, approaching the continuum limit $X_i \to \mathbb{R}^n$, it converges to the underlying continuous operator. The main use case of neural operators is to approximate the solution of partial differential equations, i.e. learn the mapping from an initial condition to the time evolved state after some time $\Delta t$ (Figure 5), but have also been applied for multi-resolution generative modelling Voleti et al. (2021); Hagemann et al. (2023). We will use them for parametrizing a flow, i.e. a vector field $\mathcal{V}_t$ connecting the free theory (base density) to the $\phi^4$-theory (target density) in a way that can be evaluated at any mesh.

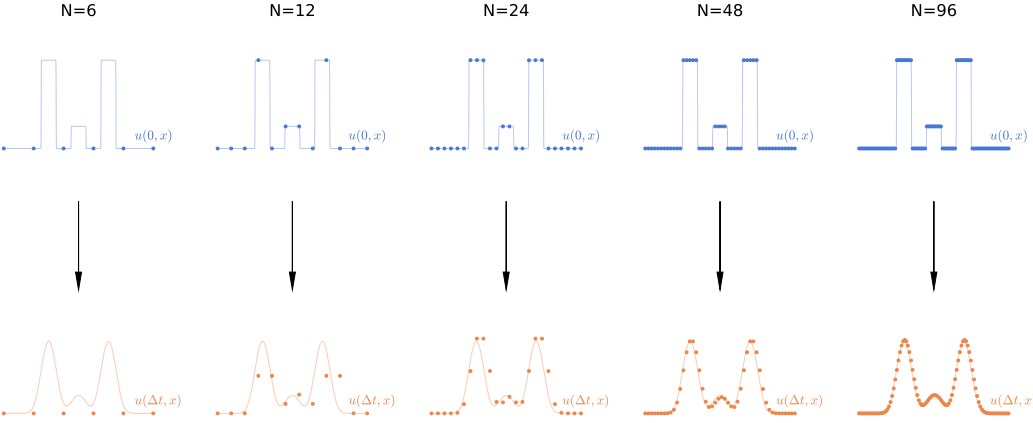

Figure 5: An operator that maps an initial condition $u(0, x)$ (top row) to its time-evolved state $u(\Delta t, x)$ (bottom row). The time evolution is given by the heat equation $\Delta u = \partial_t u$. The blue dots denote the evaluation of $u(0, x)$ on a discrete mesh, while the orange dots denote the output of the operator (a convolution in this case) evaluated on that same mesh. As the mesh gets denser, the operator becomes a better approximation of the map between the continuous $u(0, x)$ (blue curve) and $u(\Delta t, x)$ (orange curve).

ADDITIONAL FIGURES

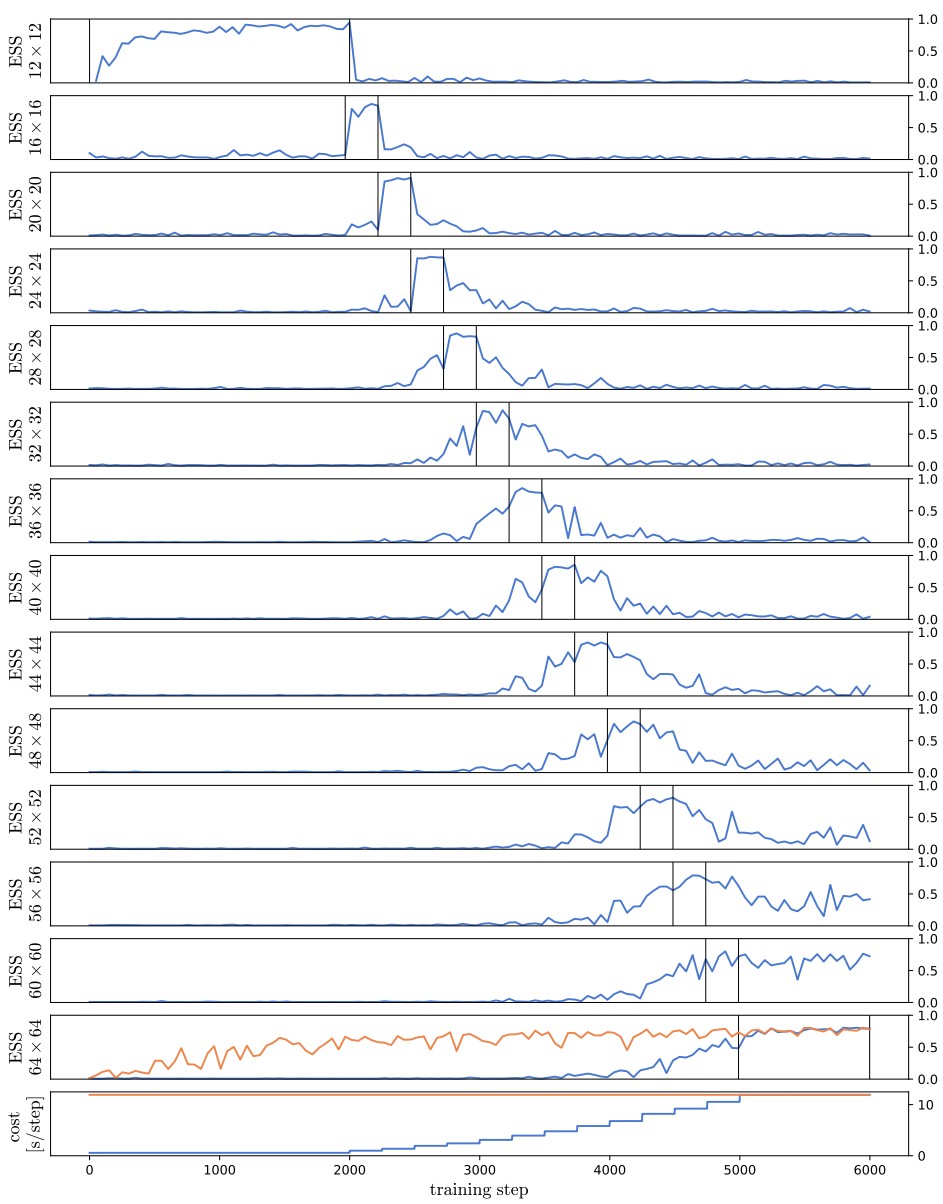

Figure 6: Experiment 4.2. ESS values computed during training from 128 samples on all the lattices the sees during training. The two thin vertical lines denote the interval during which the model is trained on the given lattice size. The orange curve corresponds to the baseline model only trained on the $64 \times 64$ lattice.

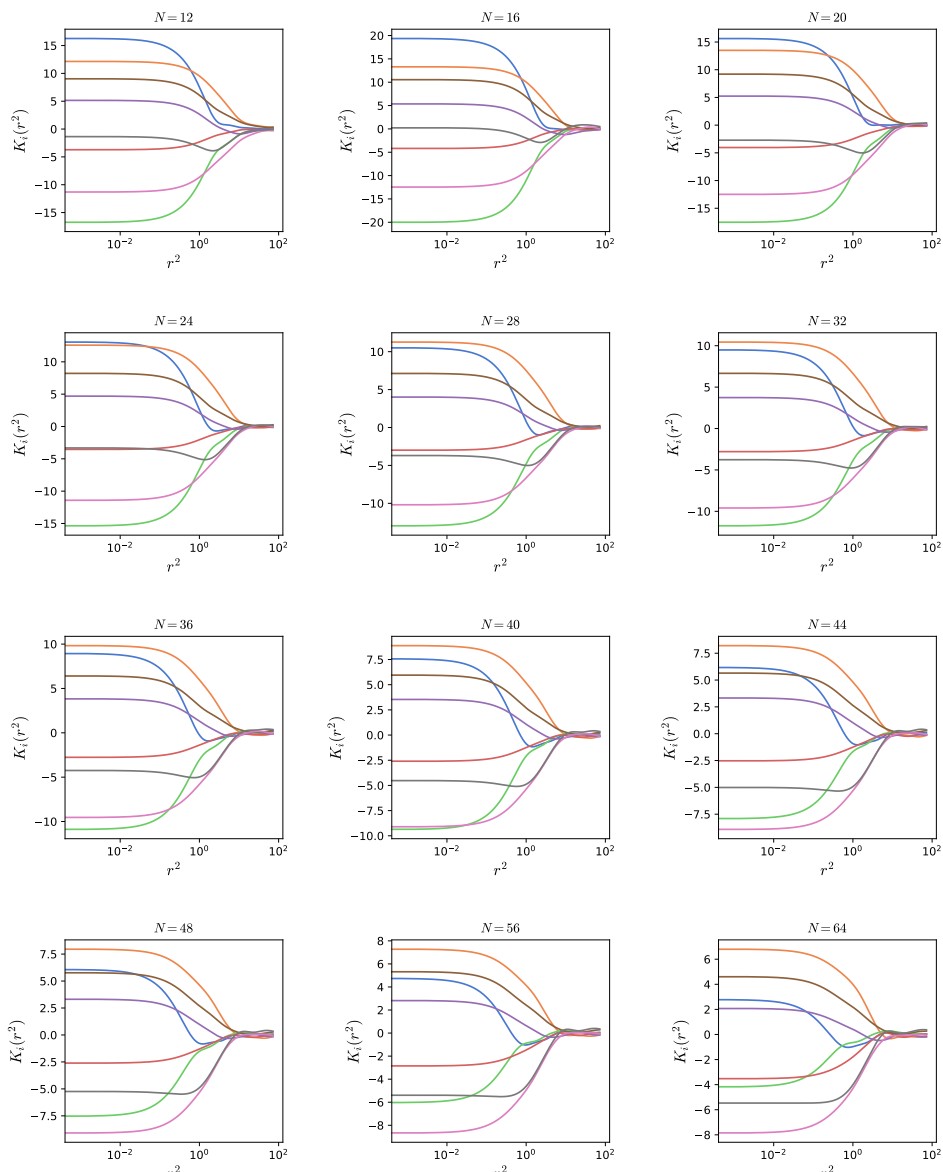

Figure 7: Experiment 4.2. Kernels (Section §3) learnt by the model on various lattice sizes.

