# OpenReview forum: "Multi-Lattice Sampling of Quantum Field Theories via Neural Operator-based Flows"
_ICLR.cc/2024/Workshop/AI4DiffEqtnsInSci — AI4DiffEqtnsInSci @ ICLR 2024 Poster_

### Official Review · Reviewer_xi3k · 2024-02-27
**The paper proposes the use of operator learning to generalize to different lattice sizes corresponding to a given quantum field theory.**

**Rating:** 6
**Confidence:** 3

**Review:**

The paper proposes to use the idea of resolution invariant learning of operators in quantum field theory, thereby enabling the use of a single model on multiple discretization (lattice) of the same underlying system.

While, the idea has been explored in many different use cases, I am unaware of it's specific use in quantum field theory.

The description of quantum field theory is adequate, given this might be new to readers of a ML venue.

However, I think the experimental section is very poorly written, and it is not easy to understand what the paper is trying to achieve in one go. I do suggest, more details on what the core experimental section is be added, plots can be moved out if needed. Without a clear message in the experimental section, the efforts made for the paper are futile.

The experimental results are somewhat convincing.

---

### Meta-Review · Area_Chair_4phu · 2024-03-01

**Recommendation:** Accept (Poster)

**Metareview:**

The work studies the application of neural operators for Quantum Field theory modeling. As mentioned by the reviewer, I also believe the experiments section needs clarification and better presentation. The authors are highly encouraged to do that for their final version. given they will improve their experiment presentation, I would be willing to accept this as poster.

---

### Decision · Program_Chairs · 2024-03-01

Accept (Poster)